# Structures of chaperone-substrate complexes docked onto the export gate in a type III secretion system

Qiong Xing [1], Ke Shi[2], Athina Portaliou[3], Paolo Rossi[1], Anastassios Economou[3] & Charalampos G. Kalodimos [1]

The flagellum and the injectisome enable bacterial locomotion and pathogenesis, respectively. These nanomachines assemble and function using a type III secretion system (T3SS). Exported proteins are delivered to the export apparatus by dedicated cytoplasmic chaperones for their transport through the membrane. The structural and mechanistic basis of this process is poorly understood. Here we report the structures of two ternary complexes among flagellar chaperones (FliT and FliS), protein substrates (the filament-capping FliD and flagellin FliC), and the export gate platform protein FlhA. The substrates do not interact directly with FlhA; however, they are required to induce a binding-competent conformation to the chaperone that exposes the recognition motif featuring a highly conserved sequence recognized by FlhA. The structural data reveal the recognition signal in a class of T3SS proteins and provide new insight into the assembly of key protein complexes at the export gate.

[1] Department of Structural Biology, St. Jude Children's Research Hospital, 263 Danny Thomas Place, Memphis, TN 38105, USA. [2] Department of Biochemistry, Molecular Biology & Biophysics, University of Minnesota, Minneapolis, MN 55455, USA. [3] Laboratory of Molecular Bacteriology, Department of Microbiology & Immunology, Rega Institute for Medical Research, Katholicke Universiteit Leuven, 3000 Leuven, Belgium. Correspondence and requests for materials should be addressed to C.G.K. (email: babis.kalodimos@stjude.org)

Type III secretion systems (T3SSs) are membrane-embedded nanomachines that export dedicated proteins from the bacterial cytoplasm[1–6]. T3SSs share the same morphology and overall structure and can be functionally classified into two classes[7,8]: the flagellar T3SS, which promotes bacterial locomotion and motility enabled by the flagellum, and the pathogenic (or non-flagellar) T3SS, which uses the injectisome to transport virulence proteins into human or animal host cells[9,10]. Both the flagellum[11–13] and the injectisome[5,6] are supramolecular complexes that are assembled by several different proteins. Flagella may also act as virulence factors because motility is crucial for the action of pathogenic bacteria[14,15]. The proteins that serve as building blocks of these organelles and the virulence proteins are typically associated with dedicated chaperones in the cytosol[16,17]. The chaperones bind and protect their cognate substrates from aggregation or premature interactions in the cytoplasm, and they assist in the targeting and delivery of the substrates at the export gate at the membrane[18–20].

The export apparatus is formed by six integral membrane proteins (in the flagellar system FlhA, FlhB, FliO, FliP, FliQ, and FliR) that are highly conserved in the flagellar and the pathogenic T3SSs[5,6]. The cytoplasmic domain of FlhA (FlhA$^C$)[21] forms the export platform onto which chaperone–substrate complexes dock to deliver the substrates for their subsequent transport to the extracellular milieu, powered primarily by the proton motive force[22,23] and assisted by the FliI ATPase[1] (Fig. 1). Crystal structures of FlhA$^C$ (refs. [24–26]) and their pathogenic T3S homologs[27,28] revealed high structural similarity. Biochemical and genetic experiments have demonstrated that FlhA is a key protein for the assembly and operation of the flagellum[21,29] and deletion of *flhA* prevents export of any flagellar protein[30]. Similar experiments in pathogenic T3SSs[31] have indicated that the FlhA homologs (referred to collectively as SctV[5,6]) are crucial for the operation of the injectisome. FlhA$^C$ assembles into a nonameric ring[28,32], which is positioned ~6 nm from the membrane surface[2] and forms a platform that operates as the export gate. Despite its central role, how the export gate recognizes and interacts with proteins in flagellar and the pathogenic T3SS is not known.

About 25 different proteins are involved in the assembly of the flagellum, which is divided into five parts from the base to the tip:

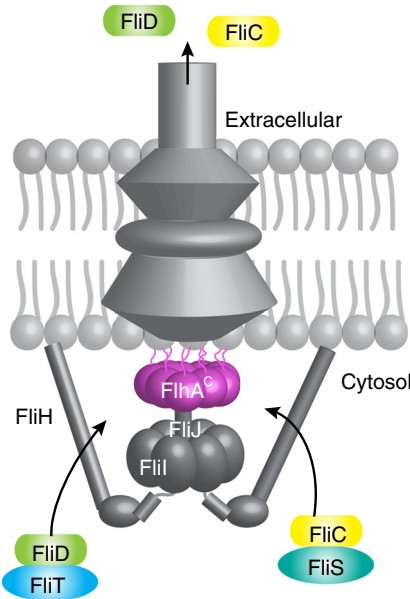

**Fig. 1** A simplified schematic of the flagellum that includes the proteins studied in this work. For detailed view, see refs. [5] [13]

the basal body, hook, hook–filament junction, filament, and filament cap[33]. The assembly of the hook and the filament is strictly sequential[13] and the final steps are controlled by the export apparatus through an elusive mechanism. The last steps of the flagellum assembly involve the export first of the filament-capping protein FliD, followed by the export of as many as 3000 flagellin (FliC) molecules, the main building blocks of the filament (Fig. 1). FliD and FliC are found in complex with their dedicated chaperones FliT and FliS, respectively, in the cytoplasm[34,35]. Biochemical and biophysical data have demonstrated that the chaperones are required for the delivery of the FliD and FliC proteins to the export gate[20,24].

Here we report the structures of the ternary complexes among the FliD and FliC flagellar proteins, their cognate chaperones FliT and FliS, and the export gate protein FlhA$^C$. The findings reveal how the export gate specifically recognizes cognate exported proteins and suggest mechanisms of operation of these protein complexes within the T3S nanomachinery.

## Results

**Interaction of FlhA with flagellar proteins.** The globular cytoplasmic domain of *Salmonella enterica* serovar Typhimurium FlhA (FlhA$^C$) encompasses residues 362–692 (36.4 kDa) and is tethered to the transmembrane domain via a linker (residues 328–362). We determined the crystal structure of FlhA$^C$ at 1.9 Å resolution (Fig. 2a, b), which shows high similarity to a previously reported structure solved at 2.8 Å resolution of a longer FlhA$^C$ construct that includes the linker (FlhA$^{C−link}$)[26] (Supplementary Fig. 1). Multiangle light scattering (MALS) data (Supplementary Fig. 2a) showed that FlhA$^C$ is monomeric in solution whereas the linker in FlhA$^{C−link}$ promotes a dimeric state that, based on the crystal structure[24,26], is mediated almost exclusively by electrostatic contacts. Indeed, FlhA$^C$ oligomerization is very sensitive to the salt concentration in solution (Supplementary Fig. 2b). Structural comparison of FlhA$^C$ and FlhA$^{C−link}$ shows that the linker does not alter the three-dimensional fold of the protein. FlhA$^C$ folds in a clamp-like structure with four discrete subdomains (d1 through d4; Fig. 2a)[24,26]. The overall topology of FlhA$^C$ and its export gate homologs from both flagellar and pathogenic T3SSs[27,28] is very similar, with the main structural variation being the opening of the clamp, as defined by the distance between the d2 and d4 domains and their relative orientation (Supplementary Fig. 1).

We used NMR (nuclear magnetic resonance) spectroscopy to obtain atomic insight into the interaction of FlhA$^C$ with cytosolic flagellar proteins. The $^1$H-$^{15}$N and $^1$H-$^{13}$C correlated spectra (Supplementary Fig. 3a, b) of FlhA$^C$ labeled in methyl-bearing amino acid residues (Ala, Ile, Met, Leu, Thr, and Val)[36–39] (Supplementary Fig. 3c) are of high quality and near-complete assignment was obtained. Addition of chaperones FliT and FliS to labeled FlhA$^C$ had no effect on the NMR spectrum of FlhA$^C$ suggesting these substrate-free chaperones do not interact with FlhA$^C$ (Supplementary Fig. 3d). This observation is in agreement with previous NMR data collected using labeled FliT chaperone and unlabeled FlhA$^C$ (ref. [20]). We also tested the interaction d1of free FliD and FliC with FlhA$^C$ by NMR and the data showed that neither of these substrates interacts with FlhA$^C$ in their chaperone-free form (Supplementary Fig. 3d).

Next, we tested the interaction between FlhA$^C$ and the chaperone–substrate complexes FliT−FliD and FliS−FliC. NMR analysis showed pronounced chemical shift perturbation on FlhA$^C$ caused by the addition of FliT−FliD or FliS−FliC (Supplementary Figs. 4a, 5a). NMR chemical shift analysis showed that the two chaperone–substrate complexes share the same binding site on FlhA$^C$, which is located in a cleft at the

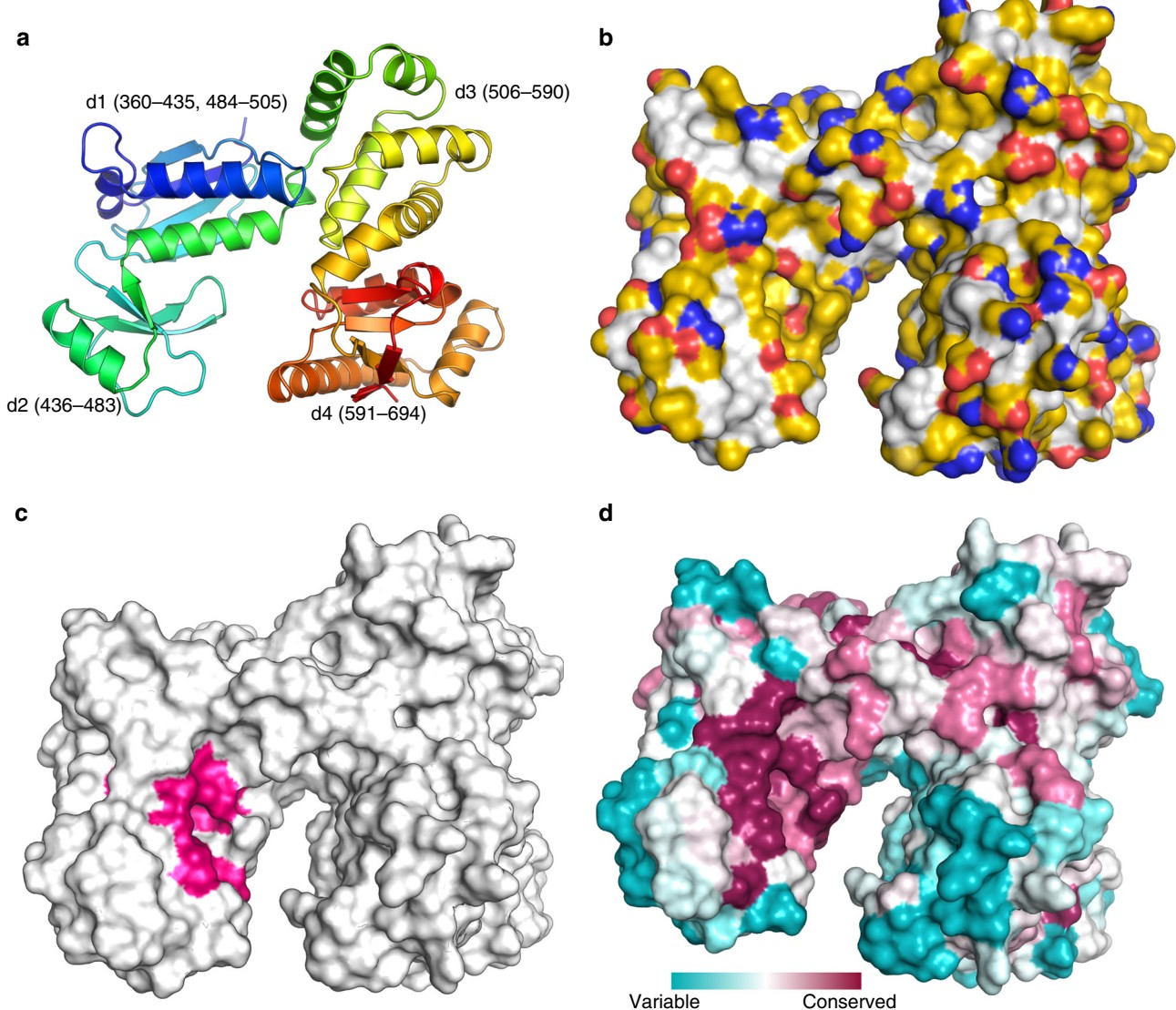

**Fig. 2** Structural properties of the export gate protein FlhA[C]. **a** Cartoon rendering of the crystal structure of FlhA[C] determined at 1.9 Å resolution. The protein is colored using a continuous-gradient color scheme from the N terminus (blue) to the C terminus (red). **b** Solvent-exposed surface rendering of FlhA[C] displayed in the YRB color scheme[59] (yellow, carbons not attached to nitrogen or oxygen; red, negatively charged; blue, positively charged). **c** The FlhA[C] site determined by NMR to interact with its binding partners is highlighted in pink. **d** Sequence conservation of FlhA[C] colored according to residue identity conservation scores obtained by ConSurf[60]. The binding site is the most conserved surface in FlhA[C]

interface between domains d1 and d2 (Fig. 2c). The location of the NMR-identified binding site is in agreement with previous biochemical data[40]. The cleft is hydrophobic and is lined by aliphatic and aromatic residues, as well as several Thr residues (Fig. 2b). The binding cleft is the most conserved surface in FlhA[C] (Fig. 2d) highlighting the importance of this region to the function of FlhA.

Analysis of the NMR data of the titration of unlabeled FlhA[C] to isotopically labeled FliT−FliD indicated that the FliT α4 helix constitutes the primary FlhA[C]-binding site (Supplementary Fig. 4b, c), confirming previous observations[20]. Similarly, NMR data showed that the N-terminal helix of FliS constitutes the primary FlhA[C]-binding site in FliS−FliC (Supplementary Fig. 5b, c). Both of these helices are part of an autoinhibitory mechanism in the free chaperones[20,41], wherein they are buried and thus unavailable for binding. Substrate (FliD and FliC, respectively) dislodges these helices in FliT and FliS, thereby activating the chaperone for binding to FlhA[C] (Supplementary Figs. 4c, 5c).

**Structure of the FlhA[C]−FliT−FliD ternary complex.** We used NMR spectroscopy to determine the structure of the FlhA[C]−FliT−FliD ternary complex in solution using NMR approaches tailored for protein complexes of large molecular weight[36,37,39]. Each one of the three proteins (FlhA[C], FliT, and FliD) was specifically labeled ([1]H and [13]C) in methyl-bearing (Ala, Ile, Leu, Met, Thr, and Val) and aromatic (Phe and Tyr) residues. In order to mitigate the severe resonance overlap, differentially labeled samples of the ternary complexes were prepared wherein typically one of the proteins was isotopically ([1]H,[13]C, and [15]N) labeled and the other two uniformly deuterated. NMR spectra of the FlhA[C]−FliT−FliD ternary complex showed significant line broadening, especially for residues located at the binding interface. Several constructs were tested and the highest quality NMR spectra suitable for solution structure determination were provided by a ternary complex formed between FlhA[C] and a fused FliT−FliD[C] construct, wherein FliD[C] consists of the last 40 C-terminal residues of FliD. Other than the superior quality of the NMR

spectra of the fused construct, due to the suppression of the unfavorable exchange processes giving rise to line broadening, the chemical shifts are essentially identical to the non-fused construct. Further NMR analysis demonstrated that FliD does not directly participate in FlhA$^C$ binding and thus a shorter FliD construct only encompassing the FliT-binding site could be used to simplify biophysical studies. The NMR observations are in line with isothermal titration calorimetry (ITC) data showing that FliD fragments longer than the FliD$^C$ have no effect on the affinity (Supplementary Fig. 4d).

The high quality spectra of the ~60-kDa FlhA$^C$−FliT−FliD$^C$ ternary complex enabled its solution structure determination by NMR. Several intermolecular NOEs were observed at the binding interface between FlhA$^C$ and FliT (Supplementary Fig. 6a) and a large number of long-distance restrains were collected using paramagnetic relaxation enhancement (PRE) experiments (see Methods) (Supplementary Fig. 6b). The structure and NMR statistics are summarized in Supplementary Table 1. The optimized protein constructs used for the NMR structure determination of the ternary complex were also used for crystallization. The ternary complex was readily crystallized and the X-ray crystallographic structure of the FlhA$^C$−FliT−FliD$^C$ ternary complex was determined at 2.75 Å resolution (Supplementary Table 2). The solution and crystal structures are essentially identical (Supplementary Fig. 6c, d).

The structure of the FlhA$^C$−FliT−FliD$^C$ complex is shown in Fig. 3a. Two distinct interfaces mediate the formation of the ternary complex. In the major interface, which buries a total surface of 350 Å$^2$, the FliT helix α4 juxtaposes with a hydrophobic cleft of FlhA$^C$ located between domains d1 and d2 (Fig. 3b). This mode of interaction is in agreement with the NMR chemical shift perturbation data and previous biochemical findings[40]. Two bulky non-polar residues, Leu102 and Tyr106, emanating from

FliT helix α4 bury their side chains into the FlhA$^C$ hydrophobic dimple. In addition to a network of intimate non-polar contacts, FliT Tyr106 forms an optimal hydrogen bond with the side chain of FlhA$^C$ Asp456 (O–H···O distance 2.3 Å). Amino acid substitutions that perturbed the binding interface decreased the ternary complex stability, with FliT residues Leu102 and Tyr106 and FlhA residues Asp456 and Leu461 being essential for complex formation (Fig. 3c). All of these residues are highly conserved (Fig. 2d and Supplementary Fig. 7).

The minor binding interface features a salt bridge between FliT Arg98 and FlhA Glu640 (Fig. 3b). Disruption of this electrostatic contact decreases the affinity 5-fold (Fig. 3c). Non-polar contacts mediated by FliT Ile93 with FlhA Met641 and Leu642 also contribute significantly to the complex stability (Fig. 3b, c). Of note, the structural data revealed that there is no direct interaction between FliD and FlhA. To maximize the binding interface with FlhAC, FliT undergoes a dramatic conformational change with helices α3 and α4 merging to one continuous, long α helix (Supplementary Fig. 4e).

**Structure of the FlhA$^C$−FliS−FliC ternary complex.** We used a similar approach to determine the structure of the FlhA$^C$−FliS−FliC ternary complex. FliS recognizes and binds the last 40 C-terminal residues of FliC[41,42] and FliS−FliC forms a stable ternary complex with FlhA$^C$ ($K_d$ ~8 μM). FliS−FliC and FliS−FliC$^C$ (where FliC$^C$ is a construct consisting of residues 454–495) have the same affinity for FlhA indicating that the first 156 residues of FliC do not contribute to FlhA binding. Similarly to FlhA$^C$−FliT−FliD, NMR analysis showed that the highest quality spectra were yielded by a ternary complex formed between FlhA$^C$ and a fused construct wherein FliC$^C$ was covalently linked to FliS. The structure of the FlhA$^C$−FliS−FliC$^C$

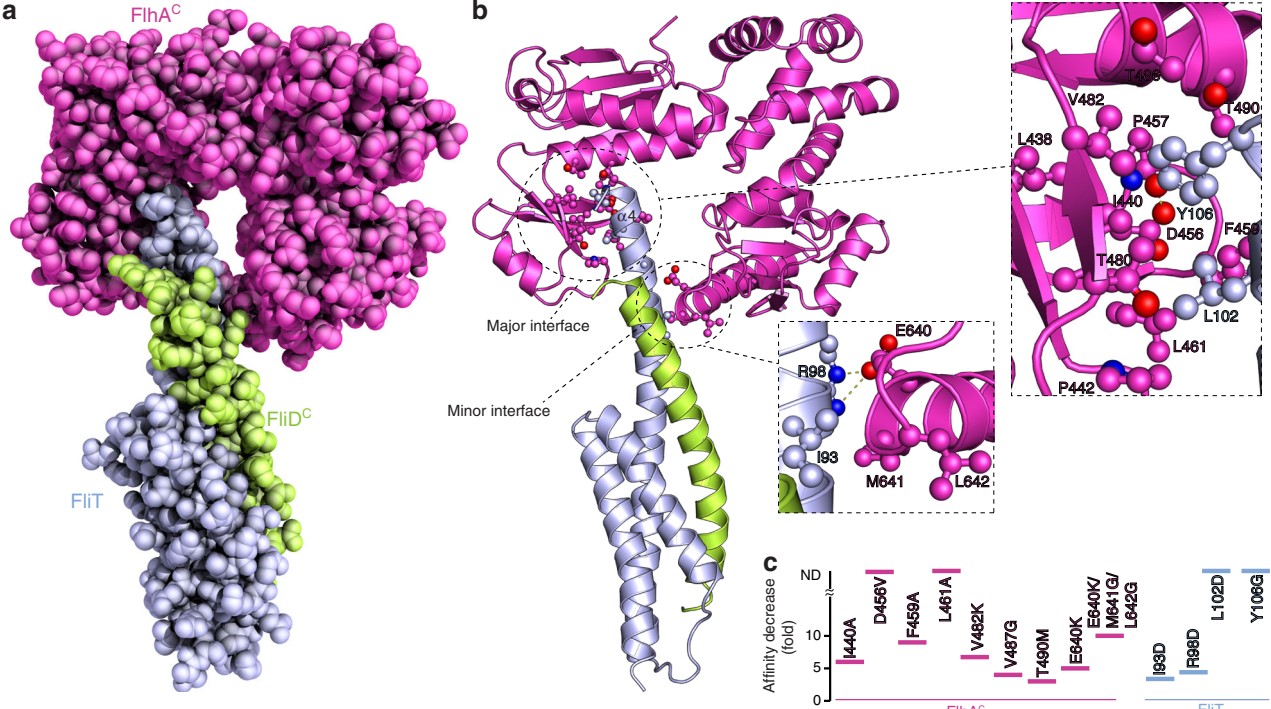

**Fig. 3** Structure of the FlhA$^C$−FliT−FliD$^C$ ternary complex. **a** Crystal structure of the ternary complex shown as a space-filling model. **b** Cartoon rendering of the structure and expanded views of the two interfaces, major and minor, between FlhA$^C$ and FliT. Residues participating in intermolecular contacts are shown as ball-and-stick. Hydrogen bonds and salt bridges are shown as broken lines. **c** Effect of the indicated amino acid substitutions on the affinity of FlhA$^C$ for FliT−FliD$^C$. The effect is given as a fold decrease relative to the affinity of the wild-type proteins ($K_d$ ~20 μM; Supplementary Fig. 4d). Non-detected binding is indicated as ND

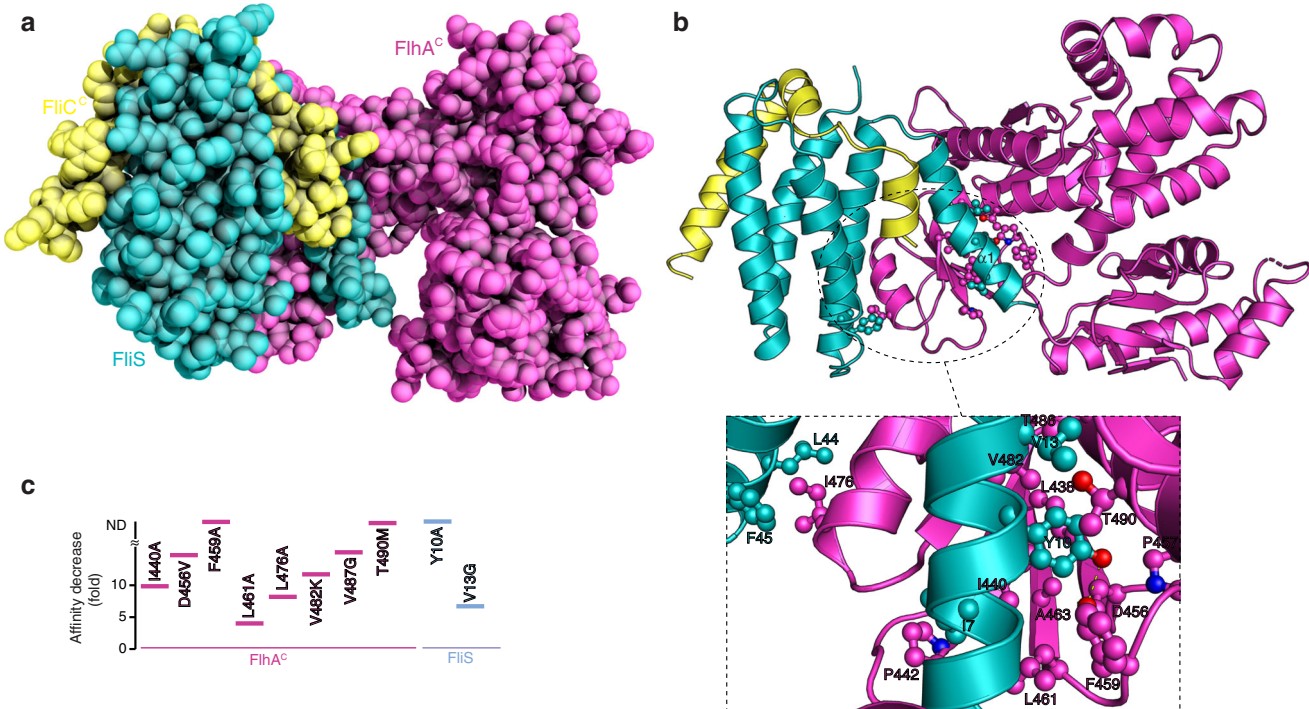

**Fig. 4** Structure of the FlhA$^C$−FliS−FliC$^C$ ternary complex. **a** Crystal structure of the ternary complex shown as a space-filling model. **b** Cartoon rendering of the structure and expanded view of the interacting regions between FlhA$^C$ and FliT. Residues participating in intermolecular contacts are shown as ball-and-stick. Hydrogen bond is shown as broken lines. **c** Effect of the indicated amino acid substitutions on the affinity of FlhA$^C$ for FliS−FliC$^C$. The effect is given as a fold decrease relative to the affinity of the wild-type proteins ($K_d$ ~10 μM; Supplementary Fig. 5d). Non-detected binding is indicated as ND

ternary complex was determined by X-ray crystallography at a resolution of 2.6 Å (Fig. 4a, b).

All contacts within the ternary complex are formed between FliS and FlhA$^C$, with FliC making no direct interaction with FlhA$^C$. Two distinct binding interfaces mediate complex formation. The most significant appears to be the juxtaposition of the FliS N-terminal helix (α1) with the FlhA$^C$ hydrophobic cleft located between the d1 and d2 domains (Fig. 4b). Thus, FliS and FliT share a common binding site on FlhA in agreement with the NMR data (Fig. 2c). Three FliS residues (Ile7, Tyr10, and Val13) emanating from helix α1 form intimate hydrophobic contacts with multiple non-polar residues within the FlhA$^C$ cleft. The binding interface buries a total surface of 680 Å$^2$. In addition, a hydrogen bond is formed between the hydroxyl group of FliS Tyr10 and FlhA$^C$ Asp456 (Fig. 4b). A minor binding interface is formed between FlhA Ile476 and FliS Leu44 and Phe45, with the three residues participating in favorable hydrophobic contacts. The structure of FlhA$^C$ in the two ternary complexes is essentially identical with a root-mean-square deviation of backbone atoms of 0.67 Å.

Amino acid substitutions that perturbed the binding interface decreased the ternary complex stability, with FliS residue Tyr10 and several FlhA residues in the binding cleft being essential for complex formation (Fig. 4c).

**Recognition mechanisms of flagellar proteins by the export gate.** Comparison of the structures of the two ternary complexes reported here illuminates the most salient features that underlie recognition of a T3S chaperone–substrate complex by the export gate (Fig. 5a). Although the FlhA-binding helix in FliT (α4) is positioned very differently along the FlhA$^C$ hydrophobic cleft than the corresponding helix in FliS (α1), their two key residues that form crucial contacts with FlhA$^C$ have the same topology. Specifically, FliS Tyr10 and FliT Tyr106 interact very similarly

with the FlhA$^C$ cleft and the Tyr residue is absolutely conserved in both FliT and FliS (Fig. 5a). A Tyr residue in this position is favored because it can form intimate hydrophobic contacts with all the non-polar residues lining the cleft, and at the same time hydrogen bond with the only charged residue in the cleft (Asp456) to neutralize its charge. The second key position is that occupied by FliS Ile7 and FliT Leu102, which form extensive, favorable hydrophobic contacts with the FlhA$^C$ cleft (Fig. 5a). The hydrophobic nature of the residue in this position is also highly conserved in both FliT and FliS (Fig. 5a).

In addition to FliD and FliC, the hook–filament junction associated proteins FlgK and FlgL are required for the assembly of the extracellular part of the flagellum. FlgK and FlgL are delivered to the export gate by their cognate chaperone FlgN. NMR analysis (Supplementary Fig. 8a–c) revealed that FlgN, with or without substrate, binds to the same cleft in FlhA$^C$ where FliT−FliD and FliS−FliC bind. Thus, FlhA uses a single binding site to engage all of the flagellar chaperones (Fig. 2c). Our NMR data indicated that the C-terminal region of FlgN is responsible for binding to FlhA. By analogy to the mode of binding of FliT and FliS, we hypothesized that Tyr122 in the C-terminal region of FlgN is the key residue for mediating complex formation with FlhA. Indeed, substitution of FlgN Tyr122[43] abolished its interaction with FlhA (Supplementary Fig. 8d), confirming the crucial role of the Tyr residue in mediating recognition between all flagellar chaperones and FlhA.

**Substrate binding to the export gate is required for motility.** To test the functional importance of the interactions we observed in the ternary complexes reported herein, we sought to determine their effect on bacterial motility. Several amino acid residues identified as crucial, based on the structural and ITC data, for complex formation were mutated and their effect on bacterial

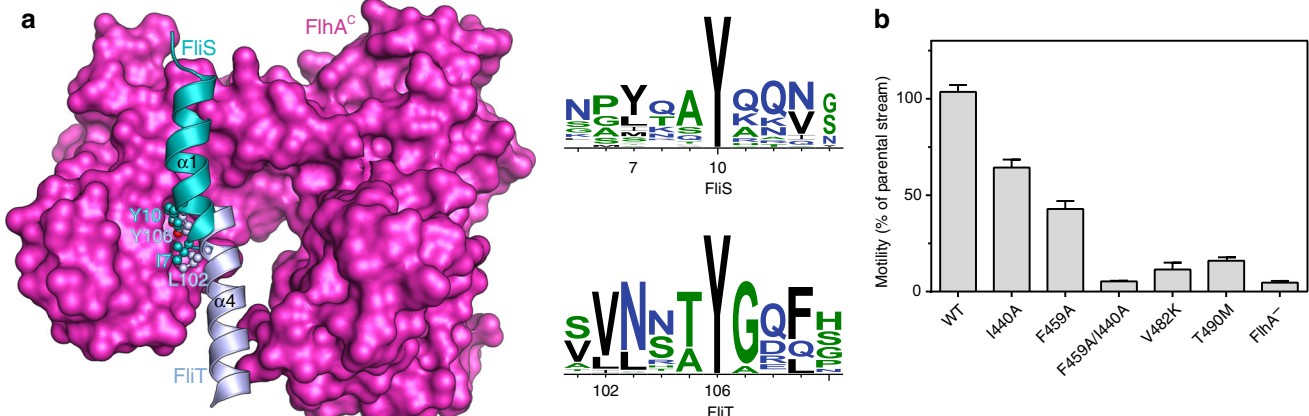

**Fig. 5** Recognition mechanisms and functional importance of the FlhA ternary complexes. **a** Superposition of the two ternary complexes (FlhA^C−FliT−FliD^C and FlhA^C−FliS−FliC^C) on FlhA^C. For clarity only the recognition helices are shown (FliT helix α4 and FliS helix α1). The two key residues recognized by FlhA is a Tyr (Tyr106 in FliT and Tyr10 in FliS) and a hydrophobic one (Leu102 in FliT and Ile7 in FliS). Sequence conservation of the recognition helices in FliT and FliS is also shown. **b** Effect of amino acid substitutions at the binding interface on the kinetics of bacterial motility. Bar graphs represent the mean value of the colony diameter and error bars represent standard deviation ($n = 6$)

motility was assessed by complementing a FlhA knockout (FlhA⁻) strain with either wild type or mutant forms of FlhA[32]. The effect was quantified by measuring the diameter of the bacteria colony in soft agar after incubation at 37 °C. FlhA mutants I440A and F459A that decrease significantly the stability of the ternary complexes (Figs. 3c and 4c) have a strong effect on motility (Fig. 5b). Overexpression of the mutated FlhA proteins restored motility, suggesting that the phenotype is due to the low affinity of the chaperone–substrate complexes for FlhA (Supplementary Fig. 9). Previous reports[43] showed that substitution of FlhA Asp456, FliS Tyr10, and FliT Tyr106 all have pronounced defect on the motility of the bacterium, in agreement with the present structural data highlighting the key role of these residues. Therefore, proper formation of the ternary complexes mediated by the binding interfaces reported here are important to the function of the flagellum.

## Discussion

Delivery of the filament-forming proteins at the export gate is an indispensable step for their export and subsequent assembly of the flagellum. Secretion of the late injectisome components and effector proteins in pathogenic bacteria employing T3SSs requires a similar targeting mechanism. While it has been known that delivery of these proteins to the export gate is assisted by dedicated chaperones, the structural basis of the process had been hitherto unknown. The present data provide atomic view of the structural features underlying the recognition mechanisms of flagellar proteins by the export gate platform. The key residue appears to be a Tyr amino acid in one of the terminal helices in chaperones, which can form extensive hydrophobic contacts with the non-polar residues lining the major binding cleft and at the same time hydrogen bond to the only charged residue that is buried in the cleft. The chaperone–substrate binding cleft is highly conserved in all flagellar systems suggesting that the recognition and targeting mechanisms are evolutionary conserved (Fig. 2d and Supplementary Fig. 10a).

The chaperone–substrate binding cleft in FlhA, which is located at the interface between the d1 and d2 domains, is not conserved in the FlhA analog (SctV) in pathogenic T3SS (Supplementary Fig. 10b). Two surfaces appear to be highly conserved in SctV: the first mediates the formation of the nonameric ring and the second one is located at the interface between the d3 and

d4 domains (Supplementary Fig. 10b). The latter can possibly serve as the chaperone–substrate binding site as mutations in this region decreased secretion[28]. This putative binding surface in SctV, in contrast to the hydrophobic nature of the binding site in FlhA, is made of highly conserved charged residues (Glu and Arg). Thus, although FlhA and SctV share very similar structures and they both bind to cognate chaperone–substrate complexes[31], the recognition motif is likely distinct in pathogenic and flagellar T3SSs. The oligomerization-mediated interface is highly conserved in SctV, but not significantly conserved in FlhA.

Interestingly, the exported proteins (e.g., FliD and FliC) do not directly interact with FlhA. Nevertheless, their binding to the chaperone is required for targeting as it poises them for binding to FlhA by relieving the autoinhibitory conformation and dislodging the recognition helix. Of note, although FliT and FliS both adopt an autoinhibitory conformation in the absence of their substrate, FlgN does not and thus is capable of binding to FlhA even in the absence of its substrates (FlgK and FlgL). Whether these different binding properties of the chaperones is physiologically significant is unclear.

A hallmark of the functionality of the flagellum and the injectisome is the hierarchical transport of proteins. In the flagellar system, FlgK and FlgL are exported first, followed by FliD and finally FliC. The mechanism underlying this process is unknown. Because the affinity of the substrate-loaded chaperones for FlhA is very similar, it is unlikely that the hierarchical transport is determined by the energetics of the various ternary complexes. The export gate platform protein forms a nonamer; hence, saturation of all nine of the binding sites would require very large differences in the relative affinities of the various chaperone–substrate complexes for the export gate.

We have fitted the structures of the ternary complexes into the cryoEM density map of the flagellum basal body[44] (Fig. 6). As noted before[44], the nonameric ring of FlhA^C fits nicely into the density map below the membrane basal base where the export gate platform is located. The bound FliT−FliD and FliS−FliC extend into the void space and these interactions bring the substrate very close to the membrane opening. How exactly the substrate dissociates from the chaperone for its export is unclear. The FliI ATPase, which is located far away from the export gate is unlikely to be the one directly disrupting the complex. The present structural data demonstrate that neither FliD nor FliC bind to FlhA. Hence, another proteinaceous factor in the

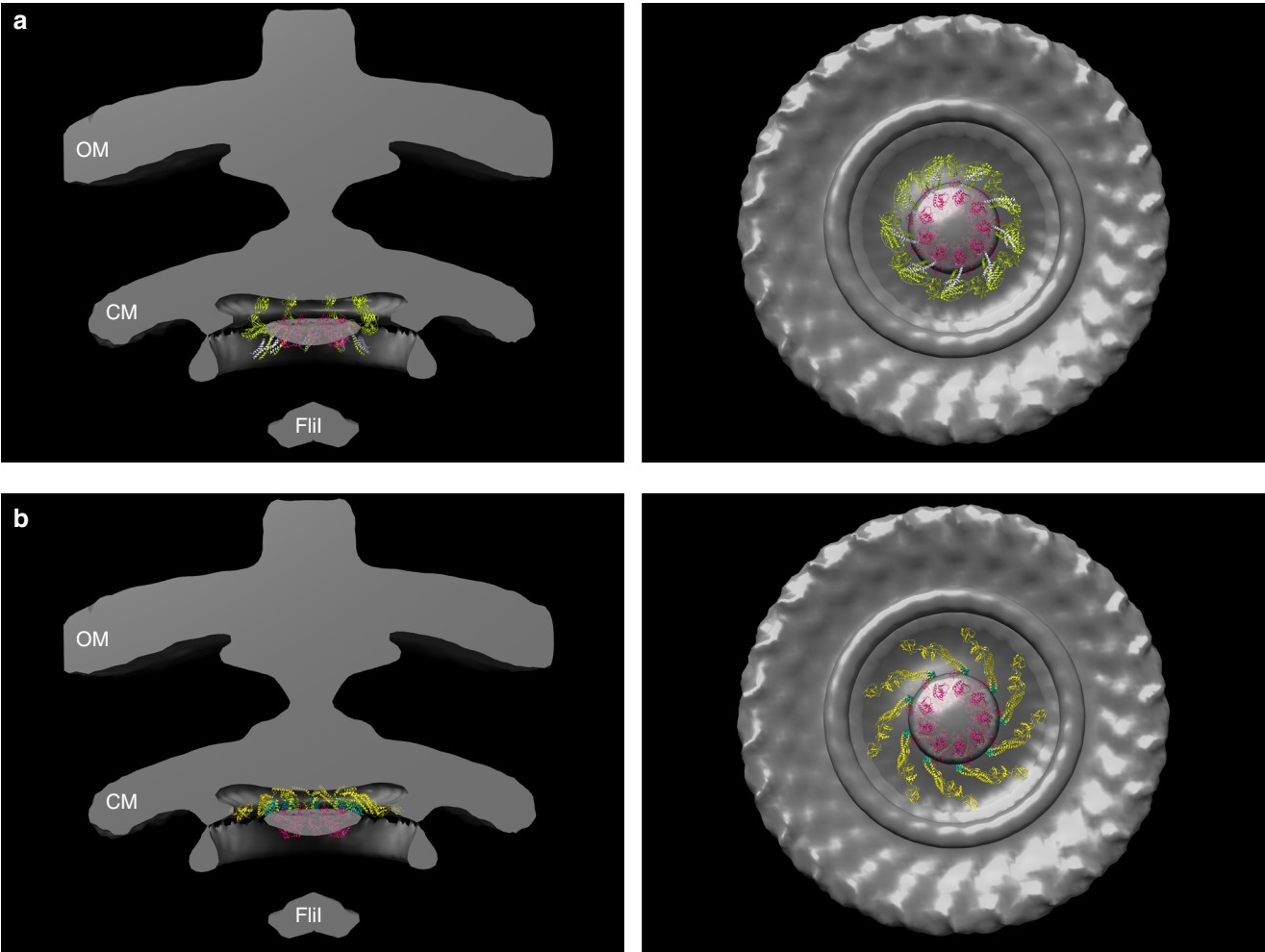

**Fig. 6** Fitting of the FlhA$^C$ ternary complexes into the cryoEM density map of the flagellum base body[44]. The nonameric oligomeric structure of FlhA$^C$ was generated using the structure of the homologous MxiA protein[28]. The ternary structures obtained in this work were then used to model the complete oligomeric nonameric ring of FlhA$^C$ bound to nine molecules of **a** FliT−FliD and **b** FliS−FliC. Non-cooperative binding of the chaperone–substrate molecules on FlhA$^C$ is assumed. The full-length FliD (5H5V) and FliC (PDB ID 3A5X) structures were used. FliI is the hexameric flagellar ATPase. CM cytoplasmic membrane; OM outer membrane. Side views of half-cut sections and bottom views are shown. The colors of the FlhA$^C$-chaperone–substrate complexes are as in Figs. 2 and 3

export apparatus should be responsible for interacting with these substrates and assist with positioning them for export.

## Methods

**Expression and preparation of proteins**. All constructs of *Salmonella enterica* serovar Typhimurium ST313 FliT, FliD, FlgN, and FlhA$^C$ were cloned into the pET16b vector (Novagen) with His$_{10}$-GB1 or His$_{10}$-MBP and tobacco etch virus (TEV) protease site in between (Supplementary Table 3). The fusion constructs were prepared by fusing FliD$^C$ to the N terminus of FliT and FliC$^C$ to the C terminus of FliS. Fusion constructs with varied linker lengths were prepared to ensure that the fusion process does not bias the conformation of the complex in any way. A 20-residue-long linker (VLFQGPSAGLVPRGSGGIEG) was selected from the pCold vector (Takara Bio). All mutants were constructed by site-directed mutagenesis using PfuTurbo high-fidelity DNA polymerase (Agilent). Unlabeled proteins were expressed in BL21(DE3) cells grown in Luria-Bertani (LB) medium in the presence of ampicillin (100 μg ml$^{-1}$) at 37 °C, and protein expression was induced at 18 °C with 0.4 mM isopropyl-β-D-1-thiogalactopyranoside (IPTG) at OD$_{600}$ ≈ 0.5 for ~48 h. Cells were harvested at OD$_{600}$ ≈ 1.5 and were suspended in lysis buffer containing 50 mM Tris-HCl (pH 7.5), 100 mM NaCl, 1% EDTA-free protease inhibitor cocktail (Sigma-Aldrich), and 20 mM imidazole. Cells were disrupted by a high-pressure homogenizer and centrifuged at 20,000 r.p.m. for 1 h. Proteins were purified using Ni Sepharose 6 Fast Flow resin (GE Healthcare), followed by tag removal by TEV protease at 4 °C for 12–20 h and gel filtration using Superdex 75 16/60. Proteins were buffer-exchanged and concentrated in Amicon filters (Millipore). Protein concentration was determined spectrophotometrically at 280 nm using the corresponding extinction coefficient.

**MALS experiments**. MALS was measured by using DAWN HELEOS-II (Wyatt Technology Corporation) downstream of a Shimadzu liquid chromatography system connected to a Superdex 200 10/300 GL (GE Healthcare) gel-filtration column. The running buffer was 50 mM NaPi (pH 6.8), 0.1 M NaCl, 0.05% NaN$_3$. Protein samples at a concentration of 0.05–0.5 mM were used. The flow rate was set to 0.5 ml min$^{-1}$ with an injection volume of 200 μl and the light scattering signal was collected at room temperature. The data were analyzed with ASTRA version 6.0.5 (Wyatt Technology Corporation).

**ITC experiments**. All ITC experiments were carried out on an iTC200 micro-calorimeter (GE Healthcare) at 25 °C. Protein samples prepared were extensively dialyzed against the ITC buffer containing 50 mM NaPi (pH 6.8), 300 mM NaCl, 0.05% NaN3. All solutions were prepared by filtering with membrane filters (pore size, 0.45 μm) and thoroughly degassing for 20 min. The sample cell (200 μL) was filled with 0.1–0.2 mM protein (FlhA$^C$ and variants), and the 60-μL injection syringe was filled with 1.0–2.0 mM protein (FliT-FiD, FliS-FliC, and variants). The titration was initiated with a preliminary 0.2-μL injection, followed by 15–25 injections of 1.9-3.9 μL, separated by a time interval of 150 s. The solution was stirred at 1000 r.p.m. Data for the first injection were discarded as it is affected by diffusion of the solution from and into the injection syringe during the initial equilibrium period. Binding isotherms profiles were generated by plotting heats of reaction normalized by the modes of injectant vs. the ratio of total injectant to total protein per injection. The data were fitted with Origin 7.0 (OriginLab Corporation) using one-site binding mode.

**Protein isotope labeling for NMR studies**. Isotopically enriched protein samples were expressed in BL21 (DE3) cells grown in minimal (M9) medium supplied with

99.9%-2H2O. Cells were induced at 18 ℃ with 0.4 mM IPTG at $OD_{600} \approx 0.4$ and typically harvested at $OD_{600} \approx 1.2$. U-[$^2$H, $^{15}$N, $^{13}$C]-labeled samples were prepared by supplementing the growth medium with $^{15}NH_4Cl$ (1 g L$^{-1}$) and $^2H_7$-$^{13}$C-glucose (2 g L$^{-1}$) and 200 μL L$^{-1}$ IsoGrow (Isotec). Methyl-protonated samples were prepared as described previously[36–39,45] using 50 mg L$^{-1}$ alpha-ketobutyric acid, 85 mg L$^{-1}$ alpha-ketoisovaleric acid, 50 mg L$^{-1}$ of $^{13}$CH3-Met, 50 mg L$^{-1}$ $^2$H$_2$, $^{13}$CH3-Ala, and 50 mg L$^{-1}$ U-$^2$H, Thr-γ2[$^{13}$CH$_3$]. For selective labeling of Phe and Tyr residues, 50 mg L$^{-1}$ of U-[$^{13}$C,$^{15}$N]-Phe and U-[$^{13}$C,$^{15}$N]-Tyr were added to the cell culture. Labeled media and compounds were purchased by Cambridge Isotope Laboratories and Isotec.

**NMR spectroscopy.** All NMR experiments were performed on Bruker AVANCE III 700, 850, and 900 MHz instruments equipped with cryogenic probes at 25 ℃. Typically, 0.3 mM isotopically labeled protein samples were prepared in 50 mM NaPi (pH 6.8), 300 mM NaCl, 0.05% NaN3, and 10% $^2$H$_2$O. All recorded spectra were processed with NMRPipe[46] and analyzed with Sparky[47]. Backbone assignment was accomplished using transverse relaxation optimized spectroscopy (TROSY)-based triple resonance experiment. $^{13}$Cα, $^{13}$Cβ, $^{13}$C′, and backbone $^1$H and $^{15}$N chemical shifts were used to compute dihedral angle (ψ, φ) restraints using TALOSN[48]. Assignment of selectively [$^1$H-methyl-$^{13}$C] labeled methyl groups was initiated with HMCM(CG)CBCA[49], and completed using a combination of 3D ($^1$H)-$^{13}$C-HMQC-NOESY-$^1$H-$^{15}$N-HMQC, 3D $^{15}$N-edited NOESY-TROSY, and ($^1$H)-$^{13}$C-HMQC-NOESY-$^1$H-$^{13}$C-HMQC experiments[50].

**PRE experiments.** PRE experiments were designed to confirm the solution structure of the ternary complex. Based on the NOE data and the binding interface we obtained, a Cys residue was introduced to three positions in FlhA$^C$ (S389C, Q473C, S637C), one position in FliT (Q84C), and one position in FliD (T441C). Protein samples with single-point cysteine substitution were expressed in M9-minimal medium with 1 g L$^{-1}$ U-$^{15}$NH4Cl, induced at $OD_{600} \approx 0.5$ with 0.4 mM IPTG at 18 ℃ for 2 days, and purified as detailed above in the presence of 5 mM reducing agent beta-mercaptoethanol (βME) in the buffer. The single cysteine mutant proteins were desalted into the reaction buffer containing 50 mM Tris (pH 6.8), 100 mM NaCl, 0.5 mM EDTA, and fivefold molar excess of N-[S-(2-pyridylthio)cysteaminyl]ethylene-diamineN,N,N′,N′-tetraacetic acid (Toronto Research Chemicals), and tenfold molar excess of divalent cation (paramagnetic: Mn$^{2+}$, diamagnetic: Ca$^{2+}$), incubating for about 24 h at 4 ℃[51]. Proteins conjugated and chelated with probes were further purified with a Mono-Q column and extensively buffer-exchanged and concentrated with Amicon filters (Millipore). Intermolecular PRE data were collected using 2D $^1$H-$^{13}$C HMQC spectra at 25 ℃ on a Bruker Avance III 850 MHz spectrometer equipped with a cryogenic probe. Resonances experiencing significant NMR signal intensity reduction (>50% intensity loss) were identified as sites being within 20 Å of the paramagnetic center, whereas residues experiencing more than 90% intensity loss were identified as sites being within 14 Å of the paramagnetic center.

**Structure determination.** The structure calculation of FlhA$^C$ in complex with FliT–FliD$^C$ was performed with CYANA 3.97 (ref. [52]), using dihedral restraints extracted as described above, NOE derived distance restraints and H-bond derived distance restraints from SO-FAST 3D ($^1$H)-$^{13}$C-HMQC-NOESY-$^1$H-$^{15}$N-HMQC, SO-FAST 3D $^{15}$N-edited NOESY-TROSY, SO-FAST $^1$H-($^{13}$C)-HMQC-NOESY-$^1$H-$^{13}$C-HMQC, and SO-FAST ($^1$H)-$^{13}$C-HMQC-NOESY-$^1$H-$^{13}$C-HMQC experiments[50] and PRE derived distance restraints from SO-FAST 2D-$^1$H-$^{13}$C-HMQC. To obtain pure intermolecular NOEs, SO-FAST ($^1$H)-$^{13}$C$_{arom}$-HMQC-NOESY-$^1$H-$^{13}$C-HMQC[50] was recorded by using U-[$^{13}$C,$^{15}$N]-Tyr-specific labeled FliT–FliD$^C$ and U-$^2$H, Ala-$^{13}$CH$_3$, Met-$^{13}$CH$_3$, Ile-δ1-$^{13}$CH$_3$, Leu/Val-$^{13}$CH$_3$/$^{12}$C$^2$H$_3$, and Thr-$^{13}$CH$_3$-labeled FlhA$^C$. Hydrogen bond restraints were obtained from analysis of NOE data and chemical shift information. Twenty structures with lowest target function were subjected to restrained molecular dynamics energy water refinement using CNS[53]. The percentage of residues falling in favored and disallowed regions of the Ramachandran plot from Procheck is: 99.7% and 0.3%, respectively. The ensemble of the 15 lowest-energy conformers are shown in Supplementary Fig. 11.

**Crystallization and data collection.** Proteins were purified and concentrated to 10 g L$^{-1}$. Crystallization was setup using the CrystalTrak system (Rigaku). A total of seven screens were setup on Intelli-Plate 96 trays (Art Robbins Instruments). Crystals were transferred into cryo-protectant containing the corresponding well solution and 20% v/v glycerol using the CryoLoop from Hampton Research, and flash frozen in liquid nitrogen. The crystals were screened at the Advanced Photon Source Northeastern Collaborative Access Team beamlines (24-ID-C). FlhA$^C$ and FlhA$^C$–FliT–FliD$^C$ crystallized in space group P1, whereas FlhA$^C$–FliS–FliC$^C$ crystallized in space group $P2_12_12_1$. X-ray diffraction data were subsequently collected and data images were processed with XDS[54]. Matthews coefficient[55] calculation indicated that there are three complexes in the asymmetric unit. Using the published FlhA$^C$ structure (PDB ID 3A5I) as a searching model, PHASER[56] located three copies of FlhA$^C$ monomers in the asymmetric unit by molecular replacement. Subsequent iterative refinement with the PHENIX suite[57], followed by model inspection/building using COOT[58] and molecular replacement using the structure of FliT−FliD$^C$ (PDB ID 5KRW)[20] generated three complete copies of the FlhA$^C$

−FliT−FliD$^C$ ternary complex, resulting in $R_{work}/R_{free}$ 22.13%/26.86%. Ramachandran analysis shows that 97.0%, 3.0%, and 0% of the protein residues are in the most favored, allowed, and disallowed region, respectively. Similar procedures were used to determine the structure of the FlhA$^C$−FliS−FliC$^C$ ternary complex. The summary of data collection and refinement statistics is shown in Supplementary Table 2. The 2Fo–Fc electron density maps are shown in Supplementary Fig. 11.

**Secretion and motility assays.** Motility assays were performed as described previously[32]. In brief, *Salmonella enterica* serovar Typhimurium Δ*flhA$^-$* strain was freshly transformed with pKG116 plasmids containing wild-type *flhA* or its mutants. Bacteria were stabbed in 0.2% LB-agar plates (1 μl from overnight pre-culture) and incubated for 6 h at 37 ℃. When indicated, plates were supplemented with 5 mM sodium salicylate to induce gene expression. Plates were scanned using Las 4000 (GE healthcare). Images were processed with Adobe Photoshop to adjust the background color and colony diameter was measured using Image Quant (GE Healthcare). Strains and plasmids were a kind gift from Marc Erhardt.

**Data availability.** Atomic coordinates for the structures have been deposited in the Protein Data Bank under accession numbers 6CH1 (FlhA$^C$), 6CH2 (FlhA$^C$−FliT−FliD$^C$ ternary complex), and 6CH3 (FlhA$^C$−FliS−FliC$^C$ ternary complex). Other data are available in this article and its Supplementary Information files, or from the corresponding author upon request.

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

## Acknowledgements

We thank Marc Erhardt for providing the Δ*flhA*-strain and the pKG116 *flhA* plasmid, and for help in setting up the motility assays. This work was supported by NIH grant AI094623 (to C.G.K.) and T3RecS (Vlaanderen Onderzoeksprojecten; #G002516N; FWO) and DIP-BiD (#AKUL/15/40—G0H2116N; Hercules/FWO) (to A.E.). This work is based upon research conducted at the Northeastern Collaborative Access Team beamlines, which are funded by the US National Institutes of Health (NIGMS P41-GM103403). The Pilatus 6 M detector on 24-ID-C beamline is funded by a NIH-ORIP HEI grant (S10 RR029205). This research used resources of the Advanced Photon Source, a U.S. Department of Energy (DOE) Office of Science User Facility operated for the DOE Office of Science by Argonne National Laboratory under Contract No. DE-AC02-06CH11357.

## Author contributions

C.G.K. conceived the study. Q.X., K.S., A.P., P.R., A.E., and C.G.K. designed the experiments and interpreted the data. Q.X. and C.G.K. wrote the paper. All authors reviewed and approved the manuscript.

## Additional information

**Competing interests:** The authors declare no competing interests.

