## [Peer Review File · Nature Communications]

Reviewers' comments:

Reviewer #1 (Remarks to the Author):

In the manuscript Kalodimos and co-workers present the structures of two ternary complexes that are crucial for motility in flagellar systems. Overall I find the work fascinating, well conducted and it moves the boundaries of the T3SS research field but is also more broadly relevant for the entire field of structural biology. The work is definitely worthy of publication in Nature Communications and I recommend acceptance after some improvement of the manuscript.

The authors has been very careful in including the relevant control experiments. For instance, it is convincingly demonstrated that FlhA only interacts complexes of FliT-FliD and Flis-FliC while no interaction is observed between FlhA and any of the free components (FliT, FliD, Flis, FliC). The NMR experiments are state of the art and these experiments has guided the researchers to constructs that could be crystallized. Finally the authors include a well-balanced structure-function analysis on basis of ITC and swarming assays. While the results are strong and convincing the manuscript would benefit by writing it a bit more general for scientists outside of the T3SS field. To this end I suggest the following changes to be considered:

(a) Add a figure that summarizes the research problem. Something similar to Fig 1 in another article from the Kalodimos group. (Khanra et al, PNAS, 2016, 113, 9798)

(b) A concluding schematic representation of the work is justified. Here it is motivated to illustrate also the activation of the chaperone by the conformational change of an inhibitory alpha helix (previous finding from the Kalodimos group). In the present manuscript the authors has discovered how this activating conformational change is harnessed in terms of function through a specific interaction with FlhA

(c) I am not sure that Fig. 5 is adding substantial information; I suggest removing this figure and replacing it with a schematic mechanistic model (see (b)).

(d) The NMR structure and the comparison with x-ray structure can be moved into the main text.

Other comments:

(1) On page 5 it is stated that FlhA-linker promotes an oligomeric state. This state is shown to be a dimer with the experiments presented in Figure S2. The word "oligomeric" is better written as "dimeric" since the data exist.

(2) The authors convincingly demonstrate that FlhA-link is an electrostatically stabilized dimer. This fits nicely with the intermolecular interactions observed for the linker in 3a5i.pdb (ref 26), a note of this convergence of data could be considered.

(3) Overall I recommend that the authors indicate the PDB codes of previously deposited structures along with the literature citations (both in the main text and the supporting information). This action enables the reader to quickly go into the pdb database for comparisons.

(4) In supplementary figure 3d the molar ratio of the various complexes and the absolute concentrations used for all components should be reported.

(5) How was the binding affinities determined for the structure-function analysis in figures 2c and 3c? If ITC was used, I suggest that the authors points to the sample ITC traces in Fig S4. If another

method (NMR?) was used, a sample experiment should be added to the supplement, preferably for the weakest binding affinity that was quantified.

(6) A cloning strategy was employed that involved covalent linkage of FliT-FliD and FliC-FliS. I suggest that the linkages are indicated in the structures in Figs 2b and 3b. The strategy is worth highlighting a bit more on page 7 since it may inspire other researchers with similar research questions.

(7) In the abstract "T3SS" and "T3S" is used, should either of the

Reviewer #2 (Remarks to the Author):

This work analyses the structure of the flagellar FlhAC target of type 3 secretion-chaperone mediated secretion as it interacts with the delivery of late flagellar secretion substrates FliD, FliC, FlgK&L by their cognate secretion chaperones FliT, FliS and FlgN, respectively.

The NMR analysis reveals interaction between chaperones and FlhAC only when bound to cognate substrates. In addition, the NMR analysis revealed specific interactions between FliT α 4 and the N-terminal helix of FliS with FlhA. These helices are buried in the non-substrate bound state and exposed when bound to cognate substrates. It's beautiful work.

The authors further determine the structures of ternary complexes and the finding that the substrate portion of the substrate-chaperone complex does not interact with FlhA provides important information related to the mechanism of secretion. This was true for both FliD-T and FliC-S complexes. These results were supported by making specific amino acid substitutions - directed by the structural analysis - which affected motility as predicted.

It was interesting to find that FlgN did not require bound secretion substrate for FlhA interaction. This could be a clue in the process of assembly as FlgK & FlgL, the FlgN cognate substrates, must be secreted and assemble prior to FliD & FliC.

This solid fundamental work that provides the foundation for structure-function analysis of secretion-chaperone mediated substrate delivery for Type 3 secretion systems.

The manuscript is very well written with an outstanding introduction.

Reviewer #3 (Remarks to the Author):

Introduction

- P3: Pathogenic T3SS should be non-flagellar or NF-T3SS as not strictly limited to pathogenic bacteria eg plant symbiosis.
- P3: "...and assisted by the FliI ATPase" should have a reference.

Results

- Reference to serotype should be consistent throughout eg *S. Typhimurium*, correct at start of results but many instances of *Salmonella enterica* alone.
- Sup Fig 1, InvA and MxiA structures possibly mixed up?
- Sup Fig 2, discrepancy between column used (Sup200 10/300) and retention volumes reported (exceed column volume), should be normalized to injection at 0 and void volume indicated.
- Sup Fig 2, highly dynamic nature of oligomerization should be acknowledged.
- ITC data eg Sup fig 4, 5, 8 should be reported as replicates (minimum triplicate) with Kd (SD).

- P8: "...first 156 residues of FliC do not contribute..." should this not be 454?
- P8-9: are there any structural differences in FlhA between FliT and FliS complexes, mention RMSD either way.
- P10: Fig4b (and Sup Fig 9), what do error bars represent?

Discussion

- P11: First paragraph would benefit from a proof read
- Discussion of the relative surface conservation patterns between FlhA and SctV (used to propose chaperone binding site isn't conserved between systems) should include that the oligomerization interface appears less conserved in FlhA as does the channel lining
- Some discussion of the different degrees of clamp opening between the various structures could be interesting, eg is opening/closing functional? or the more closed NF-T3SS homologues reflecting different substrate binding location?
- Sup fig 10b - the location of the FlhA chaperone binding cleft should be indicated
- Fig. 5, state colours to help reader distinguish FlhA vs chaperone:effector

Crystallization

- Mathew coefficient should be Matthews coefficient with reference
- Hamptonresearch should be Hampton Research
- Stated that complexes all P1 but FliS-FliC is P212121 in Table 2
- First R-free row in Table 2 should be R-free flags?

Inclusion of data seems a bit optimistic for FliS-FliC ($CC1/2 < 0.3$ in high res shell), justify or cut data.

Manuscript NCOMMS-18-00217

Referee #1

1. *“Add a figure that summarizes the research problem. Something similar to Fig 1 in another article from the Kalodimos group. (Khanra et al, PNAS, 2016, 113, 9798)”*

Response: We have added the schematic as new Fig. 1.

2. *“A concluding schematic representation of the work is justified. Here it is motivated to illustrate also the activation of the chaperone by the conformational change of an inhibitory alpha helix (previous finding from the Kalodimos group). In the present manuscript the authors has discovered how this activating conformational change is harnessed in terms of function through a specific interaction with FlhA.*

3. *“I am not sure that Fig. 5 is adding substantial information; I suggest removing this figure and replacing it with a schematic mechanistic model (see (b)).”*

Response to points 2 and 3: We prefer to keep Fig. 6 (previously Fig. 5) as it shows the real EM map and the fitting of the current structural data and thus the integration of the new ternary complexes in the context of the flagellum.

3. *“The NMR structure and the comparison with x-ray structure can be moved into the main text.”*

Response: As we noted in the main text, the NMR and X-ray structures are essentially identical. We believe that a more detailed comparison is probably of no interest to the non-specialist to justify moving it to the main text.

4. *“On page 5 it is stated that FlhA-linker promotes an oligomeric state. This state is shown to be a dimer with the experiments presented in Figure S2. The word “oligomeric” is better written as “dimeric” since the data exist.”*

Response: Text revised as suggested.

5. *“The authors convincingly demonstrate that FlhA-link is an electrostatically stabilized dimer. This fits nicely with the intermolecular interactions observed for the linker in 3a5i.pdb (ref 26), a note of this convergence of data could be considered.”*

Response: We revised the text to make it clear that the crystal structure is consistent with an electrostatically stabilized dimer.

6. *“Overall I recommend that the authors indicate the PDB codes of previously deposited structures along with the literature citations (both in the main text and the supporting information). This action enables the reader to quickly go into the pdb database for comparisons.”*

Response: All PDB codes are included in Fig. S1.

7. *“In supplementary figure 3d the molar ratio of the various complexes and the absolute concentrations used for all components should be reported.”*

Response: The information has been added in the legend.

8. "How was the binding affinities determined for the structure-function analysis in figures 2c and 3c? If ITC was used, I suggest that the authors points to the sample ITC traces in Fig S4. If another method (NMR?) was used, a sample experiment should be added to the supplement, preferably for the weakest binding affinity that was quantified."

Response: ITC was used and many of the binding isotherms are included in the SI (Figs. S4, S5, S8).

9. "A cloning strategy was employed that involved covalent linkage of *FliT-FliD* and *FliC-FliS*. I suggest that the linkages are indicated in the structures in Figs 2b and 3b. The strategy is worth highlighting a bit more on page 7 since it may inspire other researchers with similar research questions."

Response: The linkages are included in the coordinates deposited with PDB. We described more extensively this strategy, which is well known among structural biologists, in our previous work on T3SS published in PNAS 2016.

10. "In the abstract "T3SS" and "T3S" is used, should either of the"

Response: Corrected.

Referee #2

No issues raised.

Referee #3

1. "Pathogenic T3SS should be non-flagellar or NF-T3SS as not strictly limited to pathogenic bacteria eg plant symbiosis."

Response: This was added in the Introduction, alongside the term "pathogenic" to make clear that either term can be used for the same system.

2. "...and assisted by the *FliI* ATPase" should have a reference."

Response: Reference added (ref. 1).

3. "Reference to serotype should be consistent throughout eg *S. Typhimurium*, correct at start of results but many instances of *Salmonella enterica* alone."

Response: Corrected.

4. "Sup Fig 1, *InvA* and *MxiA* structures possibly mixed up?"

Response: No mix up, they are the correct ones.

5. "Sup Fig 2, discrepancy between column used (Sup200 10/300) and retention volumes reported (exceed column volume), should be normalized to injection at 0 and void volume indicated."

Response: Retention volumes were normalized and Fig. S2 was revised.

6. "Sup Fig 2, highly dynamic nature of oligomerization should be acknowledged."

Response: Legend text revised to indicate the fast equilibrium between the two states.

7. "ITC data eg Sup fig 4, 5, 8 should be reported as replicates (minimum triplicate) with K_d (SD)."

Response: All relevant figures have been updated to include the K_d with SD values from a triplicate.

8. "P8: "...first 156 residues of FliC do not contribute..." should this not be 454?"

Response: Corrected.

9. "Are there any structural differences in FlhA between FliT and FliS complexes, mention RMSD either way."

Response: The rmsd (0.67 Å) is now included on p. 9.

10. "Fig4b (and Sup Fig 9), what do error bars represent?"

Response: Bar graphs represent the mean value of the colony diameter and error bars represent standard deviation. Information added to the legend.

11. "Discussion of the relative surface conservation patterns between FlhA and SctV (used to propose chaperone binding site isn't conserved between systems) should include that the oligomerization interface appears less conserved in FlhA as does the channel lining"

Response: The following sentence was added to Discussion: The oligomerization-mediated interface is highly conserved in SctV, but not significantly conserved in FlhA.

12. "Some discussion of the different degrees of clamp opening between the various structures could be interesting, eg is opening/closing functional? or the more closed NF-T3SS homologues reflecting different substrate binding location?"

Response: This information is now included in Fig. S1.

13. "Sup fig 10b - the location of the FlhA chaperone binding cleft should be indicated"

Response: Fig. S10 has been revised to indicate the cleft.

14. "Fig 5, state colours to help reader distinguish FlhA vs chaperone:effector"

Response: Done.

15. "Stated that complexes all P1 but FliS-FliC is P212121 in Table 2"

Response: Table 2 corrected and Methods section "Crystallization and data collection" was updated.

16. "First R-free row in Table 2 should be R-free flags?"

Response: Corrected.

17. "Inclusion of data seems a bit optimistic for FliS-FliC (CC1/2<0.3 in high res shell), justify or cut data."

Response: Structure refinement was redone at 2.68 and Table S2 was updated.

18. "Matthew coefficient should be Matthews coefficient with reference".

Response: Corrected and referenced added.

19. "Hamptonresearch should be Hampton Research"

Response: Corrected.

20. "P11: First paragraph would benefit from a proof read"

Response: Done.